# Sorafenib-Regorafenib Sequential Therapy in Japanese Patients with Unresectable Hepatocellular Carcinoma—Relative Dose Intensity and Post-Regorafenib Therapies in Real World Practice

**DOI:** 10.3390/cancers11101517

**Published:** 2019-10-09

**Authors:** Wan Wang, Kaoru Tsuchiya, Masayuki Kurosaki, Yutaka Yasui, Kento Inada, Sakura Kirino, Koji Yamashita, Shuhei Sekiguchi, Yuka Hayakawa, Leona Osawa, Mao Okada, Mayu Higuchi, Kenta Takaura, Chiaki Maeyashiki, Shun Kaneko, Nobuharu Tamaki, Hiroyuki Nakanishi, Jun Itakura, Yuka Takahashi, Yasuhiro Asahina, Nobuyuki Enomoto, Namiki Izumi

**Affiliations:** 1Department of Gastroenterology and Hepatology, Musashino Red Cross Hospital, 1-26-1,Kyonan-cho, Musashino-shi, Tokyo 180-8610, Japan; w.ou@musashino.jrc.or.jp (W.W.); tsuchiya@musashino.jrc.or.jp (K.T.); kurosaki@musashino.jrc.or.jp (M.K.); yutakay@musashino.jrc.or.jp (Y.Y.); k.inada@musashino.jrc.or.jp (K.I.); sa.kirino@musashino.jrc.or.jp (S.K.); ykoji1007@yahoo.co.jp (K.Y.); s.sekiguch@musashino.jrc.or.jp (S.S.); y.hayakawa@musashino.jrc.or.jp (Y.H.); r.oosawa@musashino.jrc.or.jp (L.O.); m.okada@musashino.jrc.or.jp (M.O.); mayu.h@musashino.jrc.or.jp (M.H.); tuf029@gmail.com (K.T.); c.maeyashiki@musashino.jrc.or.jp (C.M.); skangast@tmd.ac.jp (S.K.); tamaki@musashino.jrc.or.jp (N.T.); nakanisi@musashino.jrc.or.jp (H.N.); jitakura@musashino.jrc.or.jp (J.I.); y.takahashi@musashino.jrc.or.jp (Y.T.); 2Department of Gastroenterology and Hepatology, Tokyo Medical Dental University, 1-5-45 Yushima, Bunkyo-ku, Tokyo 113-8510, Japan; asahina.gast@tmd.ac.jp; 3First Department of internal Medicine, Faculty of Medicine, University of Yamanashi, Shimokato, Chuo, Yamanashi 409-3898, Japan; enomoto@yamanashi.ac.jp

**Keywords:** hepatocellular carcinoma, sorafenib, regorafenib, sequential therapy, relative dose intensity (RDI)

## Abstract

Background: We aimed to explore the relative dose intensity (RDI) and post-regorafenib treatments in regorafenib therapy. Methods: The medical records of 38 patients treated with regorafenib between July 2017 and June 2019 at our institution were collected. The RDI of regorafenib for the first month (1M-RDI) was calculated. Results: The overall survival (OS) and progression-free survival (PFS) were 12.4 and 3.7 months. The objective response rate and disease control rate were 13.2% and 71.1%. The median total dose of regorafenib in the first month was 2080 mg (240–3360 mg), and the median 1M-RDI was 61.9% (7.1–100%). Patients with 1M-RDI ≥ 50% showed significantly longer OS and PFS than patients with 1M-RDI < 50% (HR 0.19, 95% CI 0.08–0.48, *p* = 0.0004 and HR 0.2, 95% CI 0.08–0.52, *p* = 0.0008). A 1M-RDI ≥ 50% (HR 0.18, 95% CI 0.06–0.55, *p* = 0.002) and hand–foot skin reaction (HR 0.03, 95% CI 0.008–0.16, *p* < 0.0001) were independently associated with OS. Post-regorafenib therapies were performed in 19 (86.4%) of 22 patients who had stopped regorafenib due to disease progression. Conclusion: A 1M-RDI ≥ 50% is clinically significant. Post-regorafenib therapies are commonly performed in real-world practice.

## 1. Introduction

Hepatocellular carcinoma (HCC) is one of the most common cancer types and the major cause of cancer-related deaths [1]. Sorafenib, a multikinase inhibitor, has been used as an established first-line systemic chemotherapy in the patients with unresectable hepatocellular carcinoma (u-HCC) [2,3]. Regorafenib is an oral multikinase inhibitor and the targets of regorafenib are vascular endothelial growth factor receptors (VEGFRs) 1–3, KIT, RET, RAF-1, BRAF, platelet-derived growth factor receptor, fibroblast growth factor receptor, and colony-stimulating factor 1 receptor (CSF1R) [4,5,6]. The use of regorafenib revealed improved survival in patients who showed disease progression after the administration of sorafenib in the randomized phase 3 RESORCE trial [7]. Regorafenib, cabozantinib [8], nivolumab [9], pembrolizumab [10], and ramucirumab [11,12] have been approved as a second-line systemic chemotherapy for u-HCC in the USA. In Japan, lenvatinib [13] has also been approved as a systemic agent for u-HCC since March 2018. Recently, Teufel et al. [14] reported biomarkers related to the response to regorafenib in patients who were enrolled in the RESORCE trial [7]. According to the results, baseline plasma concentrations of proteins and miRNAs had significant associations with overall survival after treatment with regorafenib. However, in real-world practice, there are still no established biomarkers in systemic therapies. Relative dose intensity (RDI) is calculated as the ratio of the actual delivered dose to the planned dose and the association between RDI and overall survival in patients with u-HCC treated with regorafenib has not yet been reported. Also, there are few reports about third-line post-regorafenib therapies for u-HCC. Therefore, we investigated patients who received regorafenib therapy in real-world practice.

## 2. Results

### 2.1. Patients

Between July 2017 and June 2019, 38 patients received regorafenib therapy at our institution. Table 1 demonstrates the baseline characteristics of these patients. Among them, three patients received regorafenib as third-line therapy for u-HCC post-treatment with lenvatinib and sorafenib. Confirmed radiological progression during sorafenib therapy and tolerability of sorafenib (sorafenib ≥ 400 mg/day for 20 or more of the last 28 days) were strongly recommended for the switch from sorafenib to regorafenib based on the inclusion criteria of the RESORCE trial [5]. In this study, 29 of 38 (76.3%) patients met the criteria, the last dose of sorafenib was less than 400 mg/day in eight patients, and one patient stopped using sorafenib due to erythema on day 9.

### 2.2. Overall Efficacy

At the end of the data cutoff (30 June 2019), the median follow-up duration was 9.3 months (1.3–21.5 months). During the observation period, 20 patients died. The median OS and PFS were 12.4 and 3.7 months, respectively (Figure 1). The median OS from the beginning of sorafenib therapy was 25.3 months. Radiological evaluation was performed in 37 patients. The objective response rate (ORR) and disease control rate in patients who received radiological evaluation by modified Response Evaluation Criteria in Solid Tumors (RECIST)were 13.2% and 71.1%, respectively. Complete response according to modified RECIST [15] was noted in zero patients, partial response (PR) in 5, stable disease (SD) in 22, and progressive disease (PD) in ten. The median treatment duration was 2.6 months. Drug discontinuation was observed in 32 patients due to disease progression (n = 22, 68.7%) or treatment-related adverse effects, AE (n = 10, 31.3%).

### 2.3. Regorafenib Treatment Regimen and Relative Dose Intensity for the First Month

The standard regimen of regorafenib is 160 mg orally once per day for 3 weeks, followed by 1 week of no treatment for each cycle. Because of the similarities of the AEs between sorafenib and regorafenib, we reduced the initial dose of regorafenib in patients who had experienced grade 2 or 3 AEs during sorafenib therapy. Child–Pugh B patients, patients with renal dysfunction, and patients with very low body weight (<50 kg) were also treated with the modified initial dose. The initial dose of regorafenib was 160 mg (n = 6), 120 mg (n = 18), 80 mg (n = 12), and 40 mg per day (n = 2). Grade 2 or 3 hand–foot skin reaction (HFSR) during sorafenib therapy was the most common reason for the initial dose modification. All of the Child–Pugh B patients (n = 5) started regorafenib therapy with the reduced dose. The median OS and PFS were 11.9 and 4.3 months, respectively, in the standard group and 13.7 and 3.9 months, respectively, in the reduced initial dose group. We further investigated the relative dose intensity (RDI) for the first month after administration of regorafenib (1M-RDI). The median total dose of regorafenib in the first month was 2080 mg (240–3360 mg), and the median 1M-RDI was 61.9% (7.1–100%). We established the cutoff values of 1M-RDI in this study at 25% intervals from a cutoff value of 100% and conducted univariate analyses of treatment results at individual points (Table 2). Patients with 1M-RDI ≥ 50% showed significantly longer OS and PFS than patients with 1M-RDI < 50% (HR 0.19, 95% CI 0.08–0.48, *p* = 0.0004 and HR 0.2, 95% CI 0.08–0.52, *p* = 0.0008) (Appendix A). Patients with 1M-RDI ≥ 50% showed significantly better pretreatment ALBI scores [16] and lower pretreatment PIVKA-II levels than patients with 1M-RDI < 50%; however, there was no significant difference in the pretreatment AFP levels between the two groups (Table 3).

### 2.4. Adverse Events Associated with Regorafenib Therapy

Adverse events led to treatment discontinuation in ten patients (31.3%), the most common being liver-associated events, including increased aspartate aminotransferase (AST) (n = 7), increased T-Bil (n = 5), and moderate ascites (n = 1). Only one patient stopped regorafenib and switched to lenvatinib due to grade 2 hand–foot skin reaction (HFSR), which continued after dose reduction and interruption of regorafenib. All 38 patients had at least one treatment-related adverse event. Table 4 shows the frequent AEs.

### 2.5. Post-Regorafenib Therapies and Efficacy of Lenvatinib as a Third-Line Agent

Thirty-two patients stopped regorafenib therapy due to disease progression (n = 22, 68.8%) or treatment-related AEs (n = 10, 31.3%), and 22 of 32 (68.8%) patients received anticancer therapy after regorafenib. As a 3rd therapy, lenvatinib therapy (n = 10), oral cytotoxic agent (n = 4), cabozantinib (n = 1), ramucirumab (n = 1), transarterial chemotherapy infusion (TAI) with cisplatin (CDDP) (n = 3), hepatic arterial infusion chemotherapy (HAIC) (n = 2) and radiation (n = 1) were performed. As a 4th therapy, lenvatinib (n = 4), oral cytotoxic agent (n = 2), HAIC (n = 1), radiation (n = 1) were performed. Lenvatinib therapy after regorafenib was performed in 14 patients, and the ORR and disease control rate (DCR) in patients who received radiological evaluation by modified RECIST were 42.9% and 85.7%, respectively. Complete response according to modified RECIST was noted in one patient, PR in five, SD in six, and PD in two. Nineteen (86.4%) of 22 patients who had stopped regorafenib due to disease progression received post-regorafenib anticancer therapy, whereas two (20%) of 10 patients who had stopped regorafenib due to AEs received post-regorafenib anticancer therapy.

### 2.6. Factors Associated with Favorable Prognosis in Regorafenib Therapy

We analyzed factors that were associated with OS in regorafenib therapy. In the univariate analyses, performance status, 1M-RDI ≥ 50%, and HFSR during regorafenib therapy were associated with OS, whereas age, body weight, pretreatment ALBI score, extrahepatic metastasis, and sorafenib duration were not significantly associated with OS. In the multivariate analysis, 1M-RDI ≥ 50% (HR 0.18, 95% CI 0.06–0.55, *p* = 0.003) and hand–foot skin reaction (HR 0.03, 95% CI 0.008–0.16, *p* < 0.0001) were independently associated with OS in regorafenib therapy (Table 5). Patients with HFSR during regorafenib showed longer survival than patients without HFSR (*p* = 0.0001, Figure 2).

## 3. Discussion

Our study was the first report about the RDI of regorafenib and the details of post-regorafenib treatments, including the efficacy of lenvatinib after regorafenib therapy. Although this report focused on a single-center study in Japan, our study contained a lot of useful data in real-world clinical practice for patients with unresectable HCC, particularly for patients who failed first-line therapy. RDI is a simple and practical tool for evaluating the total delivered dose of an anticancer agent per unit of time, expressed as a percentage of the target dose [16,17]. We found that in regorafenib therapy patients with u-HCC, the OS and PFS in patients with 1M-RDI ≥ 50% were significantly longer than those in patients with 1M-RDI < 50%. A multivariate analysis by Kawashima et al. [18] showed 1M-RDI status to be significantly associated with PFS (*p* = 0.002) in sorafenib therapy patients with first therapy-refractory metastatic renal cell carcinoma. It is well known that regorafenib has a similar chemical structure to sorafenib. The comparison between patients with 1M-RDI ≥ 50% and < 50% showed that patients with 1M-RDI ≥ 50% had significantly better pretreatment ALBI scores and lower pretreatment PIVKA-II levels. This suggests that the patients with better pretreatment ALBI scores are good candidates for regorafenib therapy, and that clinical management, including dose modification, is very important in sorafenib-regorafenib therapy.

The median OS in our study (12.4 months) was similar to that in the RESORCE trial (10.6 months) [5]; however, in two recent reports from Korea [19] and Japan [20], the median OS was not attained in the Korean report, whereas it was reported as 17.3 months in the multicenter study in Japan. The PFS in this study (3.7 months) was similar to those of the RESORCE trial (3.1 months) and the Korean report (3.7 months) but shorter than that of the Japanese multicenter study (6.9 months). There are reasons for which our results are not similar to those of the two recent reports. In comparison to the Korean reports, our patients were 13 years older (75 years versus 62 years), and the proportion of hepatitis B virus (HBV) patients was lower (18% versus 67%). The last dose of sorafenib was 800 mg/day in 48% of the patients in Korea, whereas only six (16%) patients received sorafenib at 800 mg/day in our study. The baseline characteristics of the patients were also different in our study when compared with the other Japanese multicenter study. All patients who were included in the multicenter study received sorafenib 400 mg or more for ≥20 of the last 28 days. The last dose of sorafenib was less than 400 mg/day in eight patients, and one patient stopped sorafenib treatment due to erythema on day 9 in our study.

In this study, we show the details of the post-regorafenib treatments. Thirty-two patients stopped regorafenib therapy due to disease progression (n = 22, 68.7%) or treatment-related AEs (n = 10, 31.3%), and 22 of 32 (68.8%) patients received anticancer therapy after regorafenib treatment. In the Korean report, 27 of 40 patients showed progression on regorafenib, and 8 (30%), including five who were treated with nivolumab, received subsequent systemic treatment. In the Japanese multicenter study, 37 of 44 patients discontinued regorafenib, and 23 (62.1%) received post-therapies. Kuzuya et al. [21] reported that the percentages of candidates were 57.3% for second-line therapy, 35.0% for regorafenib, and 23.3% for ramucirumab in the patients who received sorafenib at their institution. Although the patients who can receive regorafenib therapy are limited according to the inclusion criteria of the RESORCE trial, the possibility of third-line therapy (post-regorafenib therapy) is high in real-world practice. Based on the results of the REFLECT trial, lenvatinib has been approved as a first-line agent for u-HCC [13]. In Japan nivolumab, pembrolizumab, and cabozantinib are not approved as systemic therapies for HCC. Ramucirumab has been approved since Jun 2019. In these situation Japanese doctors use lenvatinib as a 3rd or 4th therapy if the patient show preserved liver function with good performance status because there are no alternative therapies. There is also no evidence about HAIC, TAI, oral cytotoxic agents and radiation after sorafenib-regorafenib sequential therapy. We carefully decided post-regorafenib therapies after discussion with the tumor board at our institution. The multicenter report [22] has shown that the OS and PFS were not significantly different between tyrosine kinase inhibitor naïve and experienced patients. In our study, the ORR and DCR in patients who received lenvatinib after regorafenib were 42.9% and 85.7%, respectively. These were similar to those obtained in the REFLECT trial.

As favorable clinical factors associated with OS in regorafenib therapy, we revealed that 1M-RDI ≥ 50% (HR 0.16, 95% CI 0.05–0.50, *p* = 0.001) and HFSR during regorafenib therapy (HR 0.06, 95% CI 0.01–0.28, *p* = 0.0004) were independently associated with OS. Previous studies [23,24,25] reported that patients with metastatic colorectal cancer who were treated with regorafenib—and who experienced HFSR—showed better OS than patients without HFSR. The correlation of HRSR and survival has also been reported in the patients who enrolled the RESORCE trial [26]. It is well known that HFSR is associated with a good prognosis in sorafenib therapy for u-HCC. Lee et al. [27] reported that differences in the incidence of HFSR would be caused by ethnic differences in genetic polymorphisms of the TNF-α, VEGF, and UGT1A9 genes. In our study, patients who experienced HFSR during sorafenib therapy showed better survival than those patients without HFSR.

The limitations of our study were a single-center study with a small number of patients and a retrospective recruitment. We included patients who did not meet the inclusion criteria of the RESORCE study. No objective pretreatment biomarkers could be found in this study. The median observation time was shorter than median OS due to the small sample size and short duration of the study period.

There is no study which compares RDI with blood concentrations in regorafenib therapy. According to the data in healthy volunteers [28], regorafenib is mainly metabolized in the liver, excreted in feces, and the results were similar among the healthy subjects. However, most HCC patients have chronic liver disease and regorafenib blood concentration would be different in these individuals. The correlation between RDI and blood concentration should be investigated and the favorable RDI based on the liver function during regorafenib therapy in a large cohort of the patients with u-HCC.

Despite a retrospective study design and a small sample size, our findings are clinically valuable and useful in the management of regorafenib therapy in real-world practice. The best cutoff value of 1M-RDI should be investigated in a large cohort study.

## 4. Methods

### 4.1. Patient and Clinical Parameters

Between July 2017 and May 2019, 38 consecutive patients with unresectable hepatocellular carcinoma received regorafenib at the Musashino Red Cross Hospital. The diagnosis of HCC was based on guidelines proposed by the Liver Cancer Study Group of Japan, the European Association for the Study of the Liver, or the American Association for the Study of Liver Diseases. In the phase 3 trial of regorafenib, tolerability for sorafenib therapy and preserved liver function were mandatory. A dose of ≥400 mg daily for at least 20 of the 28 days before discontinuation was mandatory, as well as Child–Pugh A. We also performed regorafenib therapy in some patients who did not meet the inclusion criteria because there was no alternative therapy. We included all patients who received regorafenib at our institution in this study and investigated the clinical outcomes. Written informed consent was received from all patients, and the ethics committee at the Musashino Red Cross Hospital (ethical code: 636) permitted the study in accordance with the Declaration of Helsinki. Each patient’s information was collected, which included sex, age, height, body weight, performance status accessed using the Eastern Cooperative Oncology Group scale, etiology of background liver disease, tumor status, laboratory data, and pretreatment tumor markers, including alpha-fetoprotein (AFP) and protein induced by vitamin K absence or antagonist-II. Liver function was accessed with the Child-Pugh and ALBI scores. ALBI scores were calculated using serum albumin and bilirubin values as follows: (ALBI score = (log10 bilirubin (μmol/L) × 0.66) + (albumin (g/L) × −0.085)) [29]. Clinical data after administration of regorafenib, initial dose, dose reduction, drug interruption, drug discontinuation, treatment duration, and adverse events (AEs) were also reported. The liver was examined via CT or MRI using a triphasic scanning technique. Local investigators independently evaluated the tumors as per the mRECIST. The initial tumor assessments were conducted within 8 weeks of administering regorafenib therapy. AEs were graded as per the Common Terminology Criteria for Adverse Events version 4.0. The standard regimen of regorafenib is 160 mg orally once per day for 3 weeks, followed by 1 week of no treatment for each cycle. The 1M-RDI was calculated as the ratio of the amount of an actually administered dose to the standard dose (160 mg/day × 21 days) for the first month.

### 4.2. Statistics

The OS was measured from the date of regorafenib administration to the date of death related to any cause. Patients who were lost to follow-up were censored at the last date we could confirm their survival. Patients who were alive were censored at the time of data cutoff. Progression-free survival (PFS) was measured from the date of regorafenib administration to the date of radiological tumor progression or death from any cause. Data are shown as the mean and standard deviation values. Statistical analyses were performed using Fisher’s exact test in categorical variables (gender, extrahepatic metastasis), Mann–Whitney U test in continuous variables which did not show normal distribution (albumin, T-Bil, PT, ALBI score, AFP, PIVKA-II, sorafenib duration), paired *t*-test in continuous variables which showed normal distribution (age, body weight) (Table 1 and Table 3). A Cox proportional hazard model was used to calculate the hazard ratio (HR) for univariate and multivariate analyses of factors associated with OS (Table 5) and the analysis of 1M-RDI of regorafenib (Table 2). OS and PFS were compared by log-rank test the Kaplan-Meier method. A *p*-value < 0.05 was considered as statistically significant (Figure 2). All statistical analyses were executed using Easy R (EZR), version 1.29 (Saitama Medical Center, Jichi Medical University, Saitama, Japan) 17, a graphical user interface for R (The R Foundation for Statistical Computing, Vienna, Austria).

## 5. Conclusions

Sorafenib–regorafenib sequential therapy is effective and tolerable in Japanese patients with unresectable HCC. A 1M-RDI ≥ 50% of regorafenib showed clinical importance. Post-regorafenib therapies were performed according to the current standards in real-world practice.

## Figures and Tables

**Figure 1 cancers-11-01517-f001:**
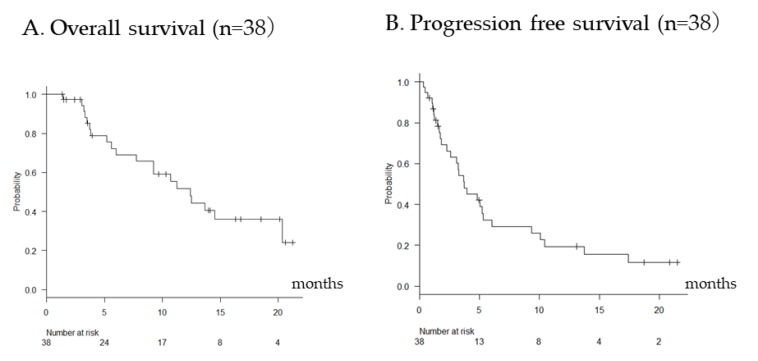
Overall survival (**A**) and progression-free survival (**B**) in all patients.

**Figure 2 cancers-11-01517-f002:**
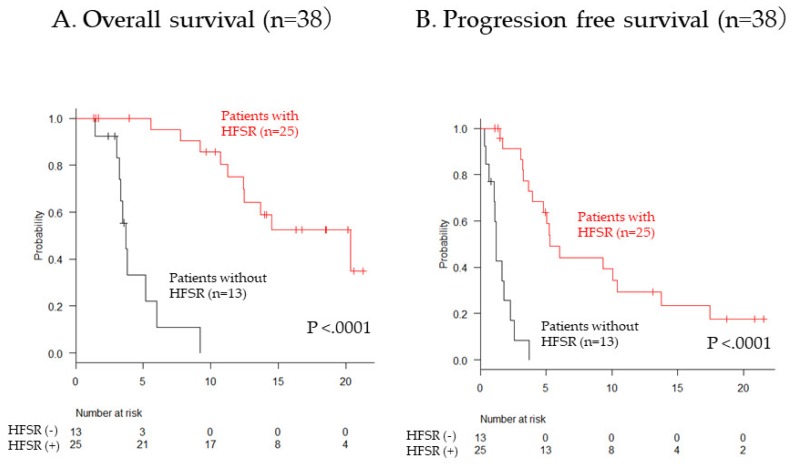
Comparison between patients with and without Hand-foot skin reaction (HFSR) in overall survival (**A**) and progression free survival (**B**).

**Table 1 cancers-11-01517-t001:** Baseline characteristics of the patients.

Factor	N = 38
Age (years), median (range)	75 (31–88)
Sex: Male/Female (%)	32 (84)/6 (16)
Body weight (kg): median (range)	57.9 (30.0–84.0)
Etiology HBV/HCV/Alcohol/Others (%)	7 (18)/16 (43)/8 (21)/7 (18)
Child–Pugh A/B/C (%)	33 (87)/5 (13)/0 (0)
Pretreatment ALBI score: median (range)	−2.33 (−1.37 to −3.76)
ECOG PS 0/1/2 (%)	17 (45)/21 (55)/0 (0)
BCLC stage A/B/C (%)	0 (0)/17 (45)/21 (55)
Major portal invasion Yes/No (%)	3 (8)/35 (92)
Baseline AFP concentration (ng/mL), median (range)	174.2 (2.6–448,620)
Baseline AFP < 400 ng/mL Yes/No (%)	22 (58)/16 (42)
Clinical course second-line/third-line (%)	35(92)/3 (8)
Sorafenib duration (months): median (range)	4.8 (0.3–62.6)
Final dose of sorafenib > 400 mg Yes/No (%)	10 (26)/28 (74)

**Table 2 cancers-11-01517-t002:** Comparison of treatment results according to 1M-RDI cutoff value.

1M-RDI Cutoff Point (%)	Patient Number	OS		
		HR	95% CI	*p*
<100 versus 100	35/3	0.87	0.20–3.77	0.85
<75 versus ≥75	23/15	0.61	0.25–1.50	0.28
<50 versus ≥50	14/24	0.19	0.08–0.48	0.0004
<25 versus ≥25	3/35	0.06	0.01–0.28	0.0002

1M-RDI: 1-month relative dose intensity; OS: overall survival; HR: hazard ratio; CI: confidence interval.

**Table 3 cancers-11-01517-t003:** Comparison of baseline characteristics between patients with 1M-RDI ≥ 50% and 1M-RDI < 50%.

Factor	1M-RDI ≥ 50%, (n = 24)	1M-RDI < 50%, (n = 14)	*p*-Value
Age (median) *	70 (31–86)	77 (58–88)	0.05
Gender (male, %)	20 (83)	12 (85)	1.00
BW (median, kg) *	58.3 (46.4–69.5)	52.7 (30.0–84.0)	0.20
Albumin (median, g/dL) *	3.5 (2.5–5.3)	3.1 (2.7–4.4)	0.06
T-Bil (median, mg/dL) *	0.6 (0.2–2.2)	0.8 (0.2–1.6)	0.24
PT (median, %) *	97 (75–117)	88 (73–113)	0.10
ALBI score (median) *	−2.4 (−3.76 to −1.41)	−2.0 (−3.03 to −1.37)	0.01
Pretreatment AFP *, (median, ng/mL)	155 (2.6–118,126)	783 (3.3–448,620)	0.54
Pretreatment PIVKA-II *, (median, mAU/mL)	306 (2.2–144,669)	5516 (11.1–668,014)	0.008
Extrahepatic metastasis, (yes, %)	2 (8.3)	1 (7.1)	1.00
Sorafenib duration *, (median, months)	5.7 (0.2–62.6)	4.1 (0.6–33.3)	0.33

*** Median (range) 1M-RDI: 1-month relative dose intensity.

**Table 4 cancers-11-01517-t004:** Adverse events (>20%).

Factor	Any n (%)	Grade ≥ 3 n (%)
HFSR	25 (65.8)	2 (5.3)
Hypertension	18 (47.4)	3 (7.9)
Diarrhea	21 (55.3)	4 (10.5)
Decreased appetite	24 (63.2)	3 (7.9)
Fatigue	28 (73.7)	1 (2.6)
Decreased body weight	16 (42.1)	0 (0)
Increased AST	20 (52.6)	2 (5.3)
Increased ALT	19 (50.0)	2 (5.3)
Increased T-Bil	11 (28.9)	0 (0)

**Table 5 cancers-11-01517-t005:** Factors associated with overall survival in regorafenib therapy.

	Univariate	Multivariate		
Factor	*p*-Value	HR	95% CI	*p*-Value
Age (years)	0.10			
Body weight (kg)	0.25			
Performance Status > 0	0.005	2.9	0.79–10.9	0.11
Pretreatment ALBI score	0.09			
Pretreatment ALT (IU/mL)	0.49			
Pretreatment AST (IU/mL)	0.25			
Pretreatment Alb (g/dL)	0.26			
Pretreatment AFP (ng/mL)	0.05			
Pretreatment PIVKA-II (mAU/mL)	0.08			
Extrahepatic metastasis	0.60			
Major portal invasion	0.25			
Sorafenib duration (months)	0.08			
HFSR during regorafenib	<0.0001	0.03	0.008–0.16	<0.0001
1M-RDI ≥ 50% of regorafenib	0.0004	0.18	0.06–0.55	0.003

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
