# Peer review of "Sorafenib-Regorafenib Sequential Therapy in Japanese Patients with Unresectable Hepatocellular Carcinoma—Relative Dose Intensity and Post-Regorafenib Therapies in Real World Practice"

_cancers, 2019, doi:10.3390/cancers11101517_

Round 1

Reviewer 1 Report

Thank you for this interesting and well-written manuscript which adds important information to the knowledge of treatment of advanced HCC. However, there are some concerns which should be addressed prior to acceptance:

Major points:

For some patients the follow-up is very short, down to 1.3 months. The median follow up was 9.3 months which is shorter than the median OS of 12.4 months. Also, a significant number of censored events can be found in the first months of the Kaplan-Meier curve. The data cutoff was June 30 2019. If longer follow-up data was available now, this would increase the significance of the data. The survival data shown in section 2.3 and figure 2 is likely underpowered with only 6 patients in the standard dose group. Thus, the statement that there is no difference might not be valid. Please review the statistic approach and in case the finding is valid, please explain. Otherwise it should not be presented. Also, please explain the rationale for only including the Child-Pugh-A patients in this analysis. In section 2.5 many different post-regorafenib therapies are mentioned. The sum of all therapies is higher than the 22 patients who received. Please explain this. Were combination therapies used? Are also later than 3rd line therapies listed? The authors conclude from the fact that ALBI-Score was better in the “high RDI”-group but not significantly associated with OS that clinical management is very important in sorafenib-regorafenib therapy. While is undoubtedly true it is problematic to draw this conclusion from the data presented as this could be just a statistical problem given the small sample size.

Minor points:

The methods section on the statistics only contains a list of the statistical tests used but not on which data which test was administered. Please elaborate on which statistical test has been used for which calculation in the methods section. Furthermore, it is useful for the reader if the test used is mentioned in the description of a specific table or figure. The correlation of HFSR and survival has also been reported from the RESORCE trial (ASCO GI 2018, Bruix et al.). This should also be discussed in the respective section. The data presented in table 2 is not clear. The formatting of the headings should be fixed. Also, please explain the number in the middle column (supposedly “Exp”?). Please explain why 1 patient was not available for radiologic evaluation. There is mistake in the numbers on page 2 line 70. According to the data PR was achieved in 5 of 37 patients and SD in 22 of 37 patients. Thus, the ORR is 13.5% (not 13.2%) and the DCR is 73.0% (not 71.0%). Also, the sum of patients adds up to 34 (5x PR, 22x SD, 7x PD), so 3 of the 37 are missing. Page 6 line 125 it should be 68.8% (not 68.7%).

Reviewer 2 Report

Wang W. propose an interesting and original work concerning clinical course of patients taking regorafenib. They show that a relative dose intensity (RDI.) impact survival. This is a simple marker, that can be used at patient bedside.

The paper is interesting, well-written and confirms previous impressions of clinical practice.

Nevertheless, some details may be cleared or improved:

In introduction, the choice of RDI  is not supported. A simple sentence may be useful. Lenvatinib choice in third line must be explained. Furthermore, some choices (f.e. arterial infusion beyond regorafenib.) seem to be not classical and some explanations may useful concerning the choice of treatment. Number of patients and a part of retrospective recruitment can appear as limitations. RDI is a simple tool. It would be interesting to compare with regorafenib blood concentrations. I think it must be discussed.

Round 2

Reviewer 1 Report

Thank you for the revised manuscript and the good work. All my points have been addressed sufficiently and the quality has improved.

Also some minor changes are proposed:

Line 76: A right parenthesis is missing. Line 97 & line 269: Figure 3 is referenced which is no longer in the manuscript. An top of page 4 the former figure 2 is still in the manuscript (but without subtitles) Line 214/215: The language of this sentence, especially the second newly added part, should be improved.

Author Response

Point by point

Reviewer 1

Thank you so much for your kind and useful comments. We rewrote our manuscript according to your comments. We sincerely appreciate your appropriate advices.

1. Line 76: A right parenthesis is missing.

Thank you so much for your kind advice. We made a new sentence with a right parenthesis as follows (page 3, line 75/76) “Drug discontinuation was observed in 32 patients due to disease progression (n = 22, 68.7%) or treatment-related adverse effects AE (n = 10, 31.3%)”.

2. Line 97 & line 269: Figure 3 is referenced which is no longer in the manuscript.

Thank you so much for your kind advice. We deleted the description about Figure 3 as follows (page 3, line 95/96 and page 8, line 269) “Patients with 1M-RDI ≥ 50% showed significantly longer OS and PFS than patients with 1M-RDI < 50% (HR 0.19, 95% CI 0.08–0.48, p = 0.0004 and HR 0.2, 95% CI 0.08–0.52, p = 0.0008)”, “OS and PFS were compared by log rank test with Kaplan-Meier method. A P-value < 0.05 was considered as statistically significant (Figure 2)”.

3. An top of page 4 the former figure 2 is still in the manuscript (but without subtitles)

Thank you so much for your kind advice. We deleted the former figure 2.

4. Line 214/215: The language of this sentence, especially the second newly added part, should be improved.

Thank you so much for your useful comment. We rewrote the sentence as follows (page 7, line 214/215) “The limitations of our study were a single-center study with a small number of patients and a part of retrospective recruitment”.